# Comparing Different Algorithms for the Pseudo-Coloring of Myocardial Perfusion Single-Photon Emission Computed Tomography Images

**DOI:** 10.3390/jimaging8120331

**Published:** 2022-12-19

**Authors:** Abdurrahim Rahimian, Mahnaz Etehadtavakol, Masoud Moslehi, Eddie Y. K. Ng

**Affiliations:** 1Department of Medical Physics, School of Medicine, Isfahan University of Medical Sciences, Isfahan 81745-33871, Iran; 2School of Mechanical and Aerospace Engineering, College of Engineering, Nanyang Technological University, 50 Nanyang Avenue, Singapore 639798, Singapore

**Keywords:** pseudo-coloring, quality assessment, quantitative indicators, SPECT image

## Abstract

Single-photon emission computed tomography (SPECT) images can significantly help physicians in diagnosing patients with coronary artery or suspected coronary artery diseases. However, these images are grayscale with qualities that are not readily visible. The objective of this study was to evaluate the effectiveness of different pseudo-coloring algorithms of myocardial perfusion SPECT images. Data were collected using a Siemens Symbia T2 dual-head SPECT/computed tomography (CT) scanner. After pseudo-coloring, the images were assessed both qualitatively and quantitatively. The qualities of different pseudo-color images were examined by three experts, while the images were evaluated quantitatively by obtaining indices such as mean squared error (MSE), peak signal-to-noise ratio (PSNR), normalized color difference (NCD), and structure similarity index metric (SSIM). The qualitative evaluation demonstrated that the warm color map (WCM), followed by the jet color map, outperformed the remaining algorithms in terms of revealing the non-visible qualities of the images. Furthermore, the quantitative evaluation results demonstrated that the WCM had the highest PSNR and SSIM but the lowest MSE. Overall, the WCM could outperform the other color maps both qualitatively and quantitatively. The novelty of this study includes comparing different pseudo-coloring methods to improve the quality of myocardial perfusion SPECT images and utilizing our collected datasets.

## 1. Introduction

Some applications of single-photon emission computed tomography (SPECT) include tumor growth estimation, genetic therapies, brain function diagnosis, and cardiovascular and coronary artery disease [1,2]. Radionuclide myocardial perfusion imaging (MPI) is a common non-invasive procedure in nuclear medicine. The SPECT-MPI method can provide valuable information about coronary perfusion and myocardial function in patients with or suspected of having coronary artery diseases [3]. In nuclear imaging, a gamma camera receives gamma rays emitted by radiopharmaceuticals. The signal then passes through internal components, including scintillation crystals, photomultiplier tubes, positioning circuits, and pulse-height analyzers. Eventually, image reconstruction algorithms are used to determine the distribution of radionuclide species. As a result of the attenuation and scattering of gamma rays, SPECT images are generally noisy [4]. Furthermore, these images are grayscale, and grayscale images have relatively low contrast between adjacent gray levels, as well as relatively few just noticeable differences (60–90 JNDs) [5]. Consequently, some of these images’ qualities are not readily visible. Pseudo-coloring images can help reveal those elements and help physicians to observe the details more clearly. Pseudo-color image processing is a method in which colors are assigned to a grayscale image based on a particular approach. The pseudo-coloring algorithm should be designed in such a way as to facilitate the accurate identification of individual data elements. In other words, the target can be distinguished from the non-target objects. In addition, the number of colors should be optimal for visual judgment [6]. Various fields benefit from pseudo-color images, including medicine and industry [7,8], since they can enhance the ability of the human visual system to perceive color.

In a study, Levin et al. (2004) [9], suggested a coloring method based on the assumption that neighboring pixels in space-time with similar intensities should have similar colors. They used a quadratic cost function and obtained an optimization problem. In their method, in order to produce a fully colorized image, a user is required to manually margin the image with a few color scratches. Moreover, Ironi et al. (2005) [10] presented a method for a grayscale image to be colorized by transferring color from a segmented example image and using a robust supervised classification scheme. Their method requires less manual effort than previous user-assisted methods.

In a study conducted by Varga and Szirányi (2016) [11], a fully automatic method was presented to colorize a grayscale image. Based on a Convolutional Neural Network, they presented a feed-forward, two-stage architecture to predict the U and V color channels. They utilized the SUN database and other images. They realized that the Quaternion Structural Similarity (QSSIM) index was a good base for quantitative evaluation. Sharma (2017) [12] developed a false-coloring method for medical-image processing and declared that the images were clear for diagnosing ligaments, tissues, and bones at a glance. Likewise, Selvapriya and Raghu (2019) [13] demonstrated that the quality and presentation of the images were improved by a pseudo-coloring process. In this study, magnetic resonance images were pseudo-colored with different algorithms such as jet color (JCM), cool color, hot color, and copper color maps. They concluded that the JCM had better performance compared to the other color maps. Whitehead et al. (2021) [14] also examined the magnetic resonance images of 26 children with sensorineural hearing loss with an average age of 7.6 ± 3 years and 13 healthy children with an average age of 7.3 ± 4 years using pseudo-color post-processing. Pseudo-color image enhancement techniques were found to improve the detection of cochlear pathology and could have therapeutic implications.

However, a few studies have evaluated the pseudo-coloring of myocardial perfusion SPECT images. Therefore, the current study aimed to evaluate the effectiveness of different pseudo-coloring algorithms.

Subsequently, the materials and methods are explained in Section 2. Section 3 presents the results and discussion, and Section 4 presents the conclusion.

## 2. Materials and Methods

First, the myocardial perfusion SPECT gray images were collected in Chamran Hospital, Iran. Second, a Wiener filter with a kernel size of 5 × 5 was used to reduce the noise of the images. Third, the images were colored using pseudo-colorization algorithms, including warm color map (WCM), hot color map (HCM), sine color map (SCM), Zahedi-proposed color map (ZCM), and jet color map (JCM). Then, the quality of the pseudo-color images was evaluated qualitatively by three experts, and quantitatively by obtaining the mean squared error (MSE), peak signal-to-noise ratio (PSNR), normalized color difference (NCD), and structure similarity index metric (SSIM) indices. Moreover, statistical analysis was performed using a one-way analysis of variance (ANOVA) test. Finally, the obtained indices were compared, and the best pseudo-coloring algorithms were selected and introduced for our dataset. Figure 1 shows the steps of this study.

### 2.1. Image Preparation and Data Collection

Our dataset was collected according to the European Association of Nuclear Medicine procedural guidelines for radionuclide MPI with SPECT and SPECT/CT [15]. Figure 2 illustrates the process of patient inclusion and exclusion. As shown, 32 patients were considered, including 17 women and 15 men. Due to pregnancy, two women were excluded from our dataset. As a result, the myocardial perfusion SPECT images from 30 patients (15 men and 17 women) in the age range of 47–75 years were included in this study. It should be mentioned that the number of images per patient was 32. These images were provided in Shahid Chamran Hospital of Isfahan, Iran, using the Siemens Symbia T2 dual-head SPECT/CT scanner with a low-energy, high-resolution collimator. A standard energy window (20%) centered at 140 keV was adjusted, and a 64 × 64 matrix size was selected as well. Sixteen views were taken for any head, and the radius of the rotation of the gamma camera head was set at 27.5 cm. The intended time for each view was 20 s. The acquisition SPECT images were reconstructed using the filtered back projection.

There were two phases in the imaging procedure, including rest and stress tests. In the rest test, based on the patient’s weight, 15–20 mCi T 99mc−MIBI (99m Tc-methoxy isobutyl isonitrile) was intravenously injected into the patient, and imaging was repeated after 45 min. The administration of drugs such as dipyridamole or Exercise (treadmill) was used to increase the patient’s heart rate, followed by the injection of 20–25 mCi T 99mc−MIBI. Then, another image was taken approximately 15–45 min later. A flowchart of the patient inclusion and exclusion is depicted in Figure 2.

### 2.2. Preprocessing (Denoising)

SPECT images are commonly noisy because gamma rays are attenuated and scattered before hitting the detector. These images are identified using low contrast and high noise levels. Therefore, images must be denoised before the pseudo-coloring process. In one study, Masoomi et al. [16], found that the Wiener filter was effective for denoising myocardial perfusion SPECT images. Therefore, this filter with kernel size 5 × 5 was applied in this study, which is defined as follows:(1)$$\mu =\frac{1}{xy}\;\underset{x,y\in \eta }{\sum }a\left(x,\;y\right)\;$$
(2)$$\;\sigma^{2}=\frac{1}{xy}\;\underset{x,y\in \eta }{\sum }a{\left(x,y\right)}^{2}-\mu^{2}$$
(3)$$F_{W}\left(x,y\right)=\mu +\frac{\sigma^{2}+\nu^{2}}{\sigma^{2}}\cdot \left(a\left(x,y\right)-\mu \right)\;$$
where *μ* and $\;\sigma^{2}\;$ represent the mean value of the pixel and the variance of the Gaussian noise in the image, respectively. Moreover, x×y and $\nu^{2}$ denote the size of the neighborhood area in kernel size *η* and the value of the noise variance in the kernel size, respectively. The average of all local variances estimated for each kernel can be used if the noise variance is not given as input [17]. The second variance adjustment was applied in the present study.

### 2.3. Pseudo-Coloring

An RGB color image is represented as an M × N × 3 array of color pixels, where each color pixel corresponds to the three values of red (R), green (G), and blue (B) components of the image at a specific spatial location. Figure 3 shows the basic principle of the pseudo-coloring process by assigning a color to each gray pixel. Each gray pixel I (*x*, *y*) is converted into three R (*x*, *y*), G (*x*, *y*), and B (*x*, *y*) values through a specific function P [I (*x*, *y*)]. Next, R (*x*, *y*), G (*x*, *y*), and B (*x*, *y*) pixels are stacked, creating a color image RGB. Differently colored images can be obtained by changing the functions P [I (*x*, *y*)].

In this study, five different pseudo-coloring algorithms, including WCM, HCM, SCM, ZCM, and JCM, were employed. Their equations are presented below in full. It is noteworthy that in all these equations, the parameter “*t*” represents the intensity value of the gray image I (*x*, *y*) after denoising.

#### 2.3.1. Warm Color Map (WCM)

In this color map, the distance between adjacent colors is understandably equal, while the color scale varies from dark blue, magenta, and orange to light yellow.
(4)$$\left(\begin{array}{c} R\left(t\right)\\ G\left(t\right)\\ B\left(t\right) \end{array}\right)=\left(\frac{1}{2\sqrt{3}}\right)\cdot \left(\begin{array}{ccc} 1+\sqrt{3} & 1-\sqrt{3} & 2\\ 1-\sqrt{3} & 1+\sqrt{3} & 2\\ -2 & -2 & 2 \end{array}\right)\cdot \left(\begin{array}{c} r\left(t\right)\sin\left(\omega t+\varphi \right)\\ r\left(t\right)\cos\left(\omega t+\varphi \right)\\ z\left(t\right) \end{array}\right)\;$$
(5)$$z\left(t\right)=\sqrt{3}\;t\;$$
(6)$$r\left(t\right)=\left\{\begin{array}{c} \sqrt{\frac{3}{2}}\;t\;\;\;\;\;\;\;\;\;\;\;0\;\leq t<\frac{1}{2}\;\\ \sqrt{\frac{3}{2}}\;\left(1-t\right)\;\;\;\;\;\;\;\;others\; \end{array}\;\right.$$

#### 2.3.2. Hot Color Map (HCM)

The most important advantage of HCM is that it clearly shows the areas with high t; therefore, the viewer is attracted to these areas. In this scale, the color slowly changes from black through red, orange, and yellow to white, all of which are classified into one color group.
(7)$$R\left(t\right)=\left\{\begin{array}{c} 2.7027\;t\;\;\;\;\;\;\;\;\;\;\;\;\;\;\;\;\;\;\;\;\;\;\;\;\;\;\;\;\;\;\;\;\;\;\;\;\;t<0.37\\ \;1\;\;\;\;\;\;\;\;\;\;\;\;\;\;\;\;\;\;\;\;\;\;\;\;\;\;\;\;\;\;\;\;\;\;\;\;\;\;\;\;\;\;\;\;\;\;\;\;t\geq 0.37 \end{array}\right.\;$$
(8)$$G\left(t\right)=\left\{\begin{array}{c} 0\;\;\;\;\;\;\;\;\;\;\;\;\;\;\;\;\;\;\;\;\;\;\;\;\;\;\;\;\;\;\;\;\;\;\;\;\;\;\;\;\;\;\;\;\;\;\;\;\;t<0.37\\ 2.7027\;t-1\;\;\;\;\;\;\;\;\;\;\;\;\;\;\;\;\;0.37\leq t\;\leq 0.74\\ 1\;\;\;\;\;\;\;\;\;\;\;\;\;\;\;\;\;\;\;\;\;\;\;\;\;\;\;\;\;\;\;\;\;\;\;\;\;\;\;\;\;\;\;\;\;\;\;\;\;t>0.74 \end{array}\right.\;$$
(9)$$B\left(t\right)=\left\{\begin{array}{c} 0\;\;\;\;\;\;\;\;\;\;\;\;\;\;\;\;\;\;\;\;\;\;\;\;\;\;\;\;\;\;\;\;\;\;\;\;\;\;\;\;\;\;\;\;\;\;\;\;t<0.74\\ 3.8461\;t-2.8461\;\;\;\;\;\;\;\;\;\;\;\;\;\;\;\;\;\;\;\;t>0.74 \end{array}\right.$$

#### 2.3.3. Sine Color Map (SCM)

This color map depicts targets with orange and red colors and non-targets as the background with a blue color.


(10)
Rt=−0.5cosπt+0.5                              Gt=sinπt                              Bt=0.5cosπt+0.5                              


#### 2.3.4. Zahedi-Proposed Color Map (ZCM)

This color map changes from blue, shades of turquoise, green, yellow, orange, and red to white.
(11)$$R\left(t\right)=\left\{\begin{array}{c} 0\;\;\;\;\;\;\;\;\;\;\;\;\;\;\;\;\;\;\;\;\;\;\;\;\;\;\;0\leq t\leq \frac{11}{20}\\ 4t-\frac{11}{5}\;\;\;\;\;\;\;\;\;\;\;\;\;\;\;\frac{11}{20}<t<\frac{4}{5}\\ 1\;\;\;\;\;\;\;\;\;\;\;\;\;\;\;\;\;\;\;\;\;\;\;\;\;\;\;\frac{4}{5}\;\leq t\;\leq 1 \end{array}\right.\;$$
(12)$$G\left(t\right)=\left\{\begin{array}{c} \begin{array}{c} 5t\;\;\;\;\;\;\;\;\;\;\;\;\;\;\;\;\;\;\;\;\;\;\;\;\;\;\;\;\;\;\;\;\;\;\;\;\;\;\;\;\;\;\;0\leq t\;\leq \frac{1}{5}\\ -\frac{20}{9}t+\frac{13}{9}\;\;\;\;\;\;\;\;\;\;\;\;\;\;\;\;\;\;\;\;\;\;\;\;\;\;\;\frac{1}{5}<t<\frac{7}{20}\\ \frac{20}{27}t+\frac{11}{27}\;\;\;\;\;\;\;\;\;\;\;\;\;\;\;\;\;\;\;\;\;\;\;\;\;\;\;\;\frac{7}{20}\;\leq t\;\leq \frac{4}{5}\\ -\frac{15}{2}t+7\;\;\;\;\;\;\;\;\;\;\;\;\;\;\;\;\;\;\;\;\;\;\;\;\;\;\;\;\;\;\;\frac{4}{5}<t<\frac{14}{15}\\ 60t-56\;\;\;\;\;\;\;\;\;\;\;\;\;\;\;\;\;\;\;\;\;\;\;\;\;\;\;\;\;\;\frac{14}{15}\;\leq t<\frac{19}{20}\\ 1\;\;\;\;\;\;\;\;\;\;\;\;\;\;\;\;\;\;\;\;\;\;\;\;\;\;\;\;\;\;\;\;\;\;\;\;\;\;\;\;\;\;\;\;\;\frac{19}{20}\leq t\;\leq 1 \end{array} \end{array}\right.\;$$
(13)$$B\left(t\right)=\left\{\begin{array}{c} 1\;\;\;\;\;\;\;\;\;\;\;\;\;\;\;\;\;\;\;\;\;\;\;\;\;\;\;\;\;\;\;\;\;\;\;\;\;\;\;\;\;\;\;\;\;\;\;\;\;\;0\leq t\;\leq \;\frac{1}{5}\\ -4t+\frac{9}{5}\;\;\;\;\;\;\;\;\;\;\;\;\;\;\;\;\;\;\;\;\;\;\;\;\;\;\;\;\;\;\;\;\;\;\;\;\;\frac{1}{5}<t<\frac{2}{5}\\ \frac{1}{5}\;\;\;\;\;\;\;\;\;\;\;\;\;\;\;\;\;\;\;\;\;\;\;\;\;\;\;\;\;\;\;\;\;\;\;\;\;\;\;\;\;\;\;\;\;\;\frac{2}{5}\;\leq t\;\leq \;\frac{14}{15}\\ 60t-56\;\;\;\;\;\;\;\;\;\;\;\;\;\;\;\;\;\;\;\;\;\;\;\;\;\;\;\;\;\;\;\;\;\;\;\frac{14}{15}<t<\frac{19}{20}\\ 1\;\;\;\;\;\;\;\;\;\;\;\;\;\;\;\;\;\;\;\;\;\;\;\;\;\;\;\;\;\;\;\;\;\;\;\;\;\;\;\;\;\;\;\;\;\;\;\frac{19}{20}\;\leq t\leq 1 \end{array}\right.\;$$


#### 2.3.5. Jet Color Map (JCM)

This color scale bar starts from blue and ends with turquoise, yellow, orange, and then red. The JCM can be considered when it is necessary to use different colors for an image.
(14)$$f\left(x\right)=\left\{\begin{array}{c} 0\;\;\;\;\;\;\;\;\;\;\;\;\;\;\;\;\;\;\;\;\;x<0\\ x\;\;\;\;\;\;\;\;\;\;\;\;\;\;0\leq x\leq 1\\ 1\;\;\;\;\;\;\;\;\;\;\;\;\;\;\;\;\;\;\;\;\;x>1 \end{array}\right.\;$$
(15)$$R\left(t\right)=f\left(1.5-\left|2\;t-1\right|\right)\;\;\;\;\;\;\;\;\;\;\;\;\;\;\;\;\;\;\;\;\;\;\;\;\;\;\;\;\;-1\leq t\leq 1\;$$
(16)$$G\left(t\right)=f\left(1.5-\left|2\;t\right|\right)\;\;\;\;\;\;\;\;\;\;\;\;\;\;\;\;\;\;\;\;\;\;\;\;\;\;\;\;\;\;\;\;\;-1\leq t\leq 1\;$$
(17)$$B\left(t\right)=f\left(1.5-\left|2\;t+1\right|\right)\;\;\;\;\;\;\;\;\;\;\;\;\;\;\;\;\;\;\;\;\;\;\;\;\;\;-1\leq t\leq 1\;$$

It is worth noting that the parameter “*t*” is between −1 and 1 in the above equations. To employ these equations, the intensity of gray images must be converted from [0 to 255] to [−1 to 1] accordingly. The following formula demonstrates the process in detail.
(18)$$I_{new}=\left(I-Min\right)\frac{newMax-newMin}{Max-Min}+newMin\to \;I_{new}=\left(I-0\right)\frac{1-\left(-1\right)}{255-0}-1=\frac{2I}{255}-1\;$$
where *I*, *Max*, and *Min* are the intensity, maximum intensity, and minimum intensity of the original image. In addition, $I_{new}$, *newMax*, and *newMin* are the intensity, maximum intensity, and minimum intensity of the new image, respectively.

### 2.4. Quantitative Indicator Calculations

Digital images are subject to a wide variety of distortions during acquisition, processing, compression, storage, transmission, and reproduction, any of which may result in a degradation of visual quality. There are many visual quality criteria to measure the quality of these images. However, the four indicators used in this study are explained in the following section.

#### 2.4.1. Mean Squared Error (MSE)

MSE measures the difference between two images in a color space. As a consequence, it is useless to calculate MSE between grayscale and color images. For this reason, each gray image was converted to an RGB image with equal values of R, G, and B. Then, the MSE was computed for the RGB and pseudo-colored images. In general, an image with a high MSE exhibits more visible distortions than an image with a low *MSE*. The *MSE* is defined as follows:(19)$$MSE=\frac{\sum_{i=1}^{M}\sum_{j=1}^{N}{\left(G_{ij}-X_{ij}\right)}^{2}}{3\;MN}\;$$

G and X represent the filtered and unfiltered images, respectively. Additionally, M and N indicate the size of the image matrix.

#### 2.4.2. Peak Signal-to-Noise Ratio (PSNR)

To determine the display quality, PSNR is calculated as the ratio of the maximum possible value of the signal to the power of noise distortion. The PSNR is usually expressed using a logarithmic decibel scale due to the wide dynamic range of many signals. The PSNR equation is as follows [18]:(20)$$PSNR=10\log_{10}\left(\frac{I_{max}{}^{2}}{MSE\left(G,X\right)}\right)\;$$

#### 2.4.3. Normalized Color Difference (NCD)

Several researchers have employed NCD as an objective measure [19]. In a particular color space (typically a laboratory color space), NCD is a measure of the distance between colors. The lower the NCD, the better the image quality. The NCD index is determined using the following formula:(21)$$NCD=\frac{\sum_{i=1}^{M}\sum_{j=1}^{N}\sqrt{\sum_{q\in \left[L,a,b\right]}{\left[G_{q}\left(i,j\right)-Cl_{q}\left(i,j\right)\right]}^{2}}}{\sum_{i=1}^{M}\sum_{j=1}^{N}\sqrt{\sum_{q\in \left[L,a,b\right]}Cl_{q}{\left(i,j\right)}^{2}}}\;$$
where $G_{q}$ and $Cl_{q}$ are gray and color images with *q* channel (in laboratory space), respectively.

#### 2.4.4. Structure Similarity Index Metric (SSIM)

SSIM index is another objective measure that is proposed in this study. It is based on how the human visual system works [20] and has been presented for analyzing the structure of an image after the coloring process. For the color image, SSIM can be separately calculated for each color channel, and then their average is computed accordingly. Different windows can be used in an image to calculate the SSIM index. The measure between two windows *x* and *y* of common size N × N is:(22)$$SSIM=\frac{\left(2\mu_{x}\mu_{y}+c_{1}\right)\left(2\sigma_{xy}+c_{2}\right)}{\left(\mu_{x}{}^{2}+\mu_{y}{}^{2}+c_{1}\right)\left(\sigma_{x}{}^{2}+\sigma_{y}{}^{2}+c_{2}\right)}\;$$
where $\mu_{x}$ and $\mu_{y}$ are the average *x* and *y*, respectively. Further, $\sigma_{x}{}^{2}$ and $\sigma_{y}{}^{2}$ represent the variance of *x* and *y*, respectively, and $\sigma_{xy}$ is the covariance of *x* and *y*.

### 2.5. Statistical Analysis

PSNR, SNR, and CNR were analyzed using IBM SPSS software, version 26. It is worth noting that the one way-ANOVA test was applied in this study with a significance level of <0.05.

### 2.6. Qualitative Assessment

A qualitative assessment was performed with trained human users. In this study, the quality of each pseudo-color image was visually evaluated by three nuclear medicine experts with many years of experience. It is noteworthy that all three experts evaluated the images under the same conditions.

## 3. Results

A myocardial perfusion SPECT image of a 61-year-old male was implemented by WCM, HCM, SCM, ZCM, and JCM pseudo-coloring algorithms, and the visual results are depicted in Figure 4.

In addition, for this case, MSE, PSNR, NCD, and SSIM indices were obtained for different pseudo-coloring algorithms and are shown in Table 1.

In addition, the graphs of the mean of MSE, PSNR, NCD, and SSIM for all cases are illustrated in Figure 5 as well as the related box plots in Figure 6. These indices were obtained using different pseudo-coloring algorithms such as WCM, HCM, SCM, ZCM, and JCM. Additionally, the related results are provided in Table 2.

## 4. Discussion

To the best of our knowledge, this is the first study for pseudo-coloring of myocardial perfusion SPECT images. Results were obtained via two approaches: (1) by qualitative assessment and (2) by quantitative assessment. In the first approach, all three experts concluded that the WCM followed by the JCM algorithms revealed the non-visible qualities of the images better than the other algorithms. In the second approach, the WCM algorithm had the highest PSNR and SSIM values. However, it had the lowest MSE values. Moreover, the SCM algorithm obtained the lowest NCD values.

As shown in Figure 5a, WCM and HCM outperformed the remaining algorithms by each achieving the lowest mean of MSE. Figure 5b also shows that WCM achieved the largest mean of PSNR compared to the other algorithms. However, based on the results shown in Figure 5c, SCM and WCM could outperform the remaining algorithms by both obtaining the lowest mean of NCD. As illustrated in Figure 5d, the largest mean of SSIM was achieved by WCM.

Based on Table 1, the WCM algorithm had the highest PSNR and SSIM values of 27.5484 ± 1.3041 (dB) and 0.9643 ± 0.0086, respectively, while it had the lowest MSE value of 0.0021 ± 0.0006. Moreover, the SCM algorithm obtained the lowest NCD value of 0.0730 ± 0.0039.

For different pseudo-coloring algorithms used in this study, the difference between MSE, PSNR, NCD, and SSIM indices was statistically significant (*p* < 0.05). However, in terms of the PSNR index, the difference between SCM and ZCM algorithms was not significantly substantial (*p* = 0.932); similarly, the difference between SCM and JCM algorithms in the SSIM index was not meaningful (*p* = 0.518). As we can see from Figure 6c, the boxplot of NCD, the only outlier data with a value of 0.061 were related to SCM.

According to Table 2, the SCM, ZCM, and JCM algorithms had high MSE values but low PSNR and SSIM values, implying that they underperformed in this regard. However, the WCM and HCM algorithms outperformed the rest by both achieving lower MSE, higher PSNR values, and SSIM values close to 1. As a result, the comparison of the results of MSE, PSNR, and SSIM indices for different algorithms used in this study revealed that the WCM algorithm was chosen quantitatively. Additionally, the WCM algorithm was also qualitatively selected by the three experts. NCD indicates the distance between colors in a specific color space; it was the best output achieved by the SCM algorithm compared to the other color-map algorithms. However, the WCM algorithm had the second best NCD after the SCM algorithm.

As mentioned earlier, the WCM algorithm performs better in both quantitative and qualitative evaluations in comparison to the other investigated algorithms. It should be noted that the second performance in quantitative and qualitative evaluations was related to HCM and JCM, respectively. The experts chose the JCM algorithm because, in their opinion, it has better contrast and visually distinguishes normal points from abnormal points more effectively compared to the other algorithms.

## 5. Conclusions

Myocardial perfusion SPECT images are usually available as grayscale images. Increasing visual clarity is beneficial to physicians for making an accurate diagnosis. Pseudo-coloring myocardial perfusion SPECT can be helpful in this regard. This study was conducted qualitatively and quantitatively to evaluate the effectiveness of pseudo-coloring algorithms such as WCM, HCM, SCM, ZCM, and JCM to increase the quality and visual clarity of myocardial perfusion SPECT. The qualitative evaluations performed by the three experts revealed that WCM and JCM algorithms both outperformed the remaining methods. Eventually, MSE, PSNR, and SSIM indices also demonstrated that the WCM algorithm could quantitatively outperform the other methods.

## Figures and Tables

**Figure 1 jimaging-08-00331-f001:**
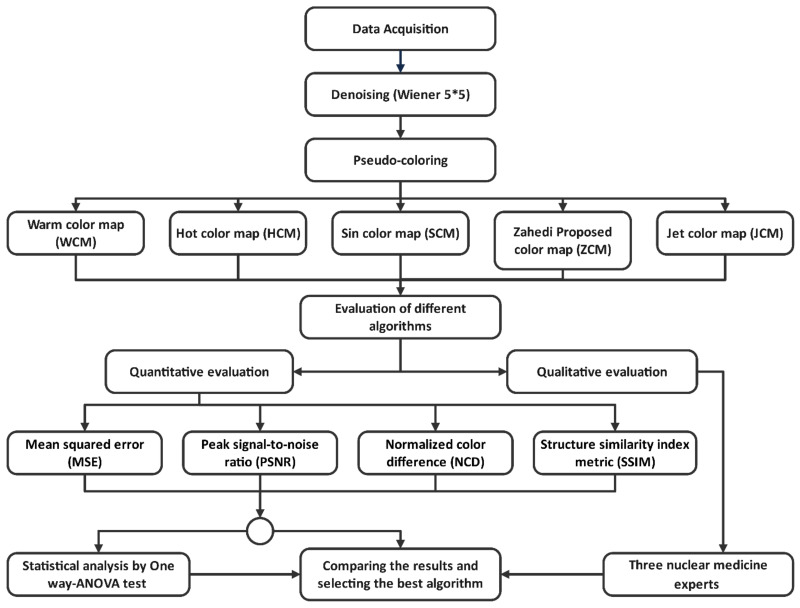
Steps of the study.

**Figure 2 jimaging-08-00331-f002:**
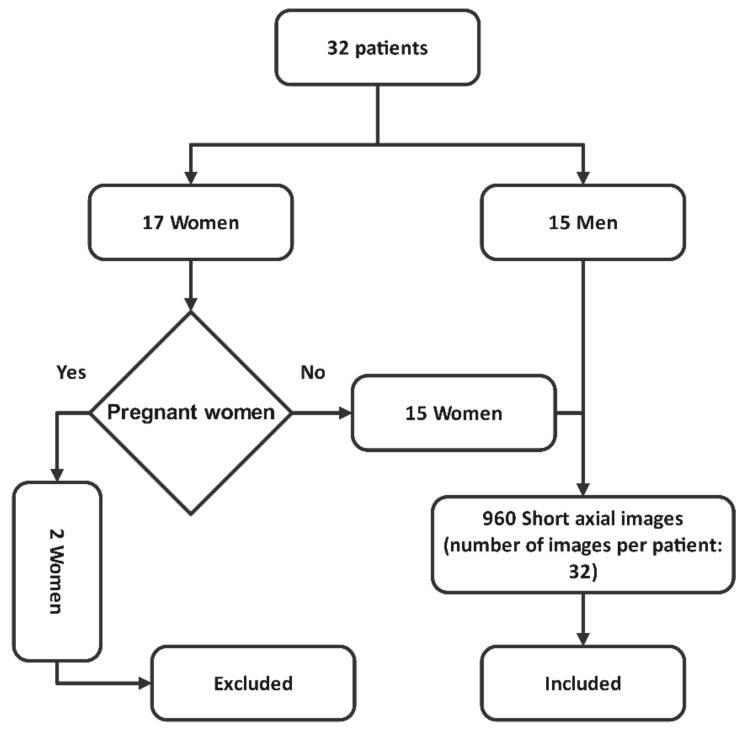
Flowchart of patient inclusion and exclusion.

**Figure 3 jimaging-08-00331-f003:**
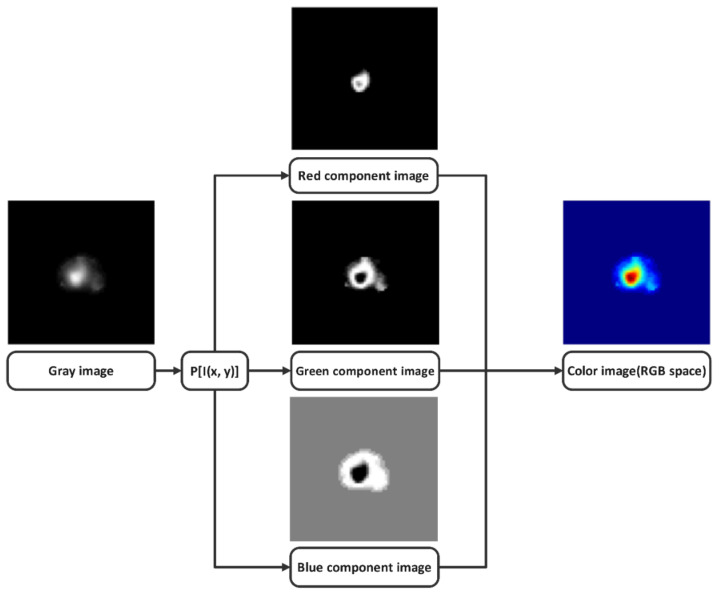
The basic principle of the pseudo-coloring process.

**Figure 4 jimaging-08-00331-f004:**
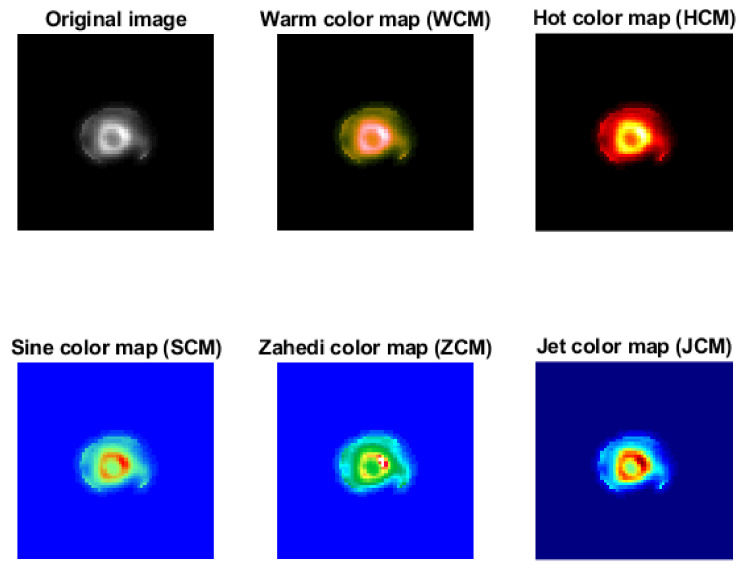
The visual results of WCM, HCM, SCM, ZCM, and JCM pseudo-coloring algorithms for a 61-year-old male. *Note*: WCM: Warm color map; HCM: Hot color map; SCM: Sine color map; ZCM: Zahedi-proposed color map; JCM: Jet color map.

**Figure 5 jimaging-08-00331-f005:**
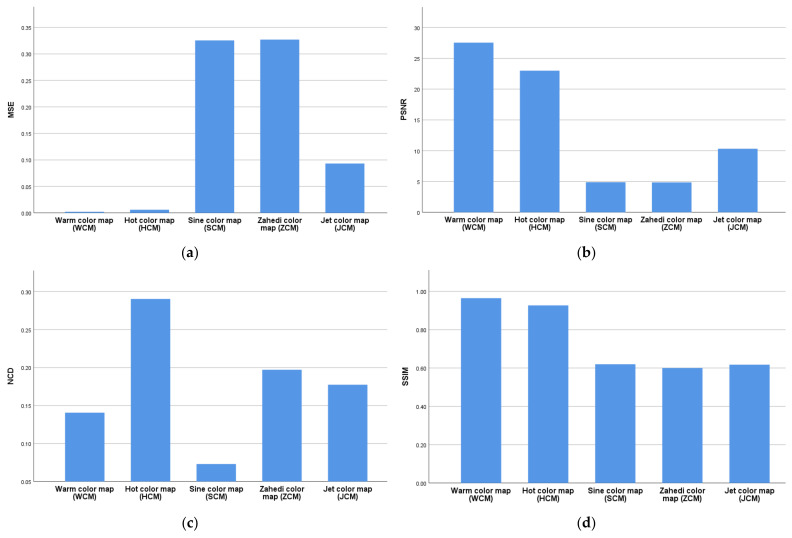
The graphs of the mean of (**a**) MSE, (**b**) PSNR, (**c**) NCD, and (**d**) SSIM. *Note*. MSE: Mean squared error; PSNR: Peak signal-to-noise ratio; NCD: Normalized color difference; SSIM: Structure similarity index metric.

**Figure 6 jimaging-08-00331-f006:**
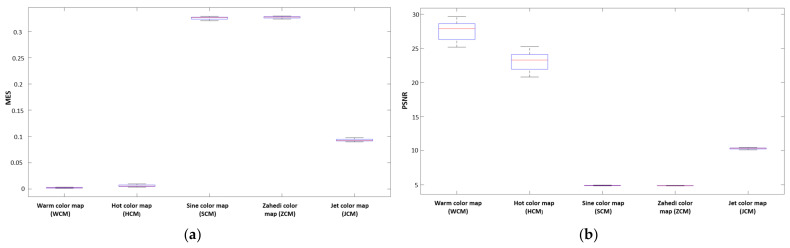
The box plots of (**a**) MSE, (**b**) PSNR, (**c**) NCD, and (**d**) SSIM. *Note*. MSE: Mean squared error; PSNR: Peak signal-to-noise ratio; NCD: Normalized color difference; SSIM: Structure similarity index metric.

**Table 1 jimaging-08-00331-t001:** Obtained MSE, PSNR, NCD, and SSIM indices by applying different pseudo-coloring algorithms for the myocardial perfusion SPECT image of a 61-year-old male. The up arrow (↑) indicates that the larger the average value, the better the model performance. The down arrow (↓) has the opposite meaning.

Pseudo-Coloring/Index	MSE (↓)	PSNR (dB) (↑)	NCD (↓)	SSIM (↑)
Warm color map (WCM)	**0.0027**	**26.3055**	0.1584	**0.9577**
Hot color map (HCM)	0.0075	21.9380	0.3284	0.9147
Sine color map (SCM)	0.3234	4.9026	**0.0732**	0.6139
Zahedi-proposed color map (ZCM)	0.3256	4.8732	0.2232	0.5906
Jet color map (JCM)	0.0938	10.2779	0.1914	0.6115

**Table 2 jimaging-08-00331-t002:** Average (± standard deviation (SD)) indices of mean squared error (MSE), peak signal-to-noise (PSNR), normalized color difference (NCD), and structure similarity index metric (SSIM) for different applied pseudo-coloring algorithms in this study. The up arrow (↑) indicates that the larger the average value, the better the model performance. The down arrow (↓) has the opposite meaning.

Pseudo-Coloring/Index	MSE ± SD (↓)	PSNR (dB) ± SD (↑)	NCD ± SD (↓)	SSIM ± SD (↑)
Warm color map (WCM)	**0.0021 ± 0.0006**	**27.5484 ± 1.3041**	0.1406 ± 0.0089	**0.9643 ± 0.0086**
Hot color map (HCM)	0.0060 ± 0.0017	22.9967 ± 1.2585	0.2903 ± 0.0232	0.9268 ± 0.0170
Sine color map (SCM)	0.3254 ± 0.0023	4.8754 ± 0.0313	**0.0730 ± 0.0039**	0.6196 ± 0.0102
Zahedi-proposed color map (ZCM)	0.3269 ± 0.0015	4.8551 ± 0.0208	0.1971 ± 0.0154	0.6001 ± 0.0142
Jet color map (JCM)	0.0931 ± 0.0022	10.3115 ± 0.1015	0.1774 ± 0.0092	0.6175 ± 0.0116

## Data Availability

Data will be provided if required.

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
