# Peer review of "Comparing Different Algorithms for the Pseudo-Coloring of Myocardial Perfusion Single-Photon Emission Computed Tomography Images"

_2313-433X, 2022, doi:10.3390/jimaging8120331_

Round 1

Reviewer 1 Report

The authors use existing methods on a new aspect of gray-level noncolored medical images extracted from spect images. even though the coloring methods are not new but the results are important.

sincerly yours

Author Response

Reviewer 1 has no specific comments.

We would like to thank you for your thoughtful comments and efforts towards improving our manuscript. Your comments provided valuable insights to refine its contents and analysis. Track changes are used in the revised manuscript to facilitate your review process. Moreover, added paragraphs are highlighted in yellow.

Reviewer 2 Report

Although the topic of this manuscript is very interesting, the authors should improve this manuscript.

a) The authors should format the equations of the manuscript more carefully and aesthetically. 

b) The authors do not provide too much related work in the introduction section. However, natural image colorization was a hot research topic. For example, deep learning based colorization: Fully automatic image colorization based on Convolutional Neural Network (2016), optimization based colorization: Colorization using optimization (2004), example based colorization: Colorization by example (2005).

c) The authors do not provide enough quantitative or qualitative results. For example, the authors could publish original gray-scale images and the different pseudo-colored images which were obtained by the examined methods.

d) The authors could visualize the obtained SSIM, PSNR, MSE values using box plots: https://www.mathworks.com/help/stats/boxplot.html

e) The authors should also polish a little bit further the English of the manuscript. 

Author Response

pls see attached file

Round 2

Reviewer 2 Report

The manuscript can be accepted now.